# Genomic Prediction of Growth Traits in Yorkshire Pigs of Different Reference Group Sizes Using Different Estimated Breeding Value Models

**DOI:** 10.3390/ani14071098

**Published:** 2024-04-04

**Authors:** Chang Yin, Haoran Shi, Peng Zhou, Yuwei Wang, Xuzhe Tao, Zongjun Yin, Xiaodong Zhang, Yang Liu

**Affiliations:** 1Department of Animal Genetics and Breeding, College of Animal Science and Technology, Nanjing Agricultural University, Nanjing 210095, China; 2021105019@stu.njau.edu.cn (C.Y.); 2023105022@stu.njau.edu.cn (H.S.); 2022105032@stu.njau.edu.cn (P.Z.); 2022805132@stu.njau.edu.cn (Y.W.); 2023805132@stu.njau.edu.cn (X.T.); 2College of Animal Science and Technology, Anhui Agricultural University, Hefei 230036, China; yinzongjun@ahau.edu.cn (Z.Y.); xdzhang1983@163.com (X.Z.)

**Keywords:** growth traits, genomic selection, multi-population, best single-step genomic linear unbiased prediction

## Abstract

**Simple Summary:**

This study addresses a major challenge in small- to medium-scale pig farming by tackling the issue of limited reference population data in breeding programs. To enhance the accuracy of genomic estimated breeding values, this research explores the benefits of combining reference populations of varying sizes. Specifically, focusing on Yorkshire pigs, this study examined the impact of different population combinations on the accuracy of genomic selection for key traits related to the growth and lean meat percentage. The findings reveal that predicting a population using data from two other populations significantly improves accuracy, offering a promising strategy for small- and medium-sized pig herds. This innovative approach has the potential to enhance genomic selection accuracy, providing valuable insights for pig farmers facing resource constraints. Ultimately, this study underscores the importance of incorporating population combinations in genetic models for predicting breeding values, contributing to more efficient and effective pig farming practices.

**Abstract:**

The need for sufficient reference population data poses a significant challenge in breeding programs aimed at improving pig farming on a small to medium scale. To overcome this hurdle, investigating the advantages of combing reference populations of varying sizes is crucial for enhancing the accuracy of the genomic estimated breeding value (GEBV). Genomic selection (GS) in populations with limited reference data can be optimized by combining populations of the same breed or related breeds. This study focused on understanding the effect of combing different reference group sizes on the accuracy of GS for determining the growth effectiveness and percentage of lean meat in Yorkshire pigs. Specifically, our study investigated two important traits: the age at 100 kg live weight (AGE100) and the backfat thickness at 100 kg live weight (BF100). This research assessed the efficiency of genomic prediction (GP) using different GEBV models across three Yorkshire populations with varying genetic backgrounds. The GeneSeek 50K GGP porcine high-density array was used for genotyping. A total of 2295 Yorkshire pigs were included, representing three Yorkshire pig populations with different genetic backgrounds—295 from Danish (small) lines from Huaibei City, Anhui Province, 500 from Canadian (medium) lines from Lixin County, Anhui Province, and 1500 from American (large) lines from Shanghai. To evaluate the impact of different population combination scenarios on the GS accuracy, three approaches were explored: (1) combining all three populations for prediction, (2) combining two populations to predict the third, and (3) predicting each population independently. Five GEBV models, including three Bayesian models (BayesA, BayesB, and BayesC), the genomic best linear unbiased prediction (GBLUP) model, and single-step GBLUP (ssGBLUP) were implemented through 20 repetitions of five-fold cross-validation (CV). The results indicate that predicting one target population using the other two populations yielded the highest accuracy, providing a novel approach for improving the genomic selection accuracy in Yorkshire pigs. In this study, it was found that using different populations of the same breed to predict small- and medium-sized herds might be effective in improving the GEBV. This investigation highlights the significance of incorporating population combinations in genetic models for predicting the breeding value, particularly for pig farmers confronted with resource limitations.

## 1. Introduction

The use of genomic selection (GS) has led to the transformation of breeding practices across animal and plant industries, which rely on factors such as reference population size, marker density, and trait heritability for precision [1,2,3]. High-precision genomic predictions for select candidates necessitate a substantial reference population comprising individuals with both genotypic and phenotypic data [2]. Nevertheless, collecting such extensive reference populations proves challenging, particularly with limited animal resources. An alternative strategy involves combining populations of different breeds to enhance the precision of genomic prediction [4,5]. Previous studies [6] have found that the Bayesian model accuracy for merged populations with different populations of the same species in simulated data is better than the GBLUP and ssGBLUP models’ accuracies. This study was conducted to verify whether the conclusions were consistent with the simulated data using real data.

While prior investigations on genomic selection in pigs have focused primarily on similar or closely related genetic backgrounds [7], limited research has been conducted on genomic selection in pigs utilizing populations characterized by diverse genetic backgrounds. It has been shown that crossing populations does not improve the accuracies of genome predictions because differences in allele substitution effects between populations reduce the accuracies of the genome predictions across populations [8]. Additionally, compared to those among dairy cattle, the consistency of linkage disequilibrium patterns among different pig populations is lower, prompting questions about the substantial improvement in dairy cattle. Traits crucial for genomic selection in pigs, such as feed intake, carcass characteristics, and meat quality, pose challenges in establishing sufficiently large reference populations. Hence, it is imperative to investigate the potential benefits of amalgamating reference groups of diverse sizes to augment the accuracy of genomic estimated breeding values in pigs.

The Yorkshire pig breed, acclaimed for its traits, such as effective utilization of feed, fast growth, valuable slaughter characteristics, and favorable lean meat composition, holds prominence in pig production. Two critical traits, namely, age at 100 kg live weight (AGE100) and backfat thickness at 100 kg (BF100), are of particular significance, with AGE100 serving as a pivotal genetic marker for the rate of growth [9] and BF100 being crucial for assessing the lean meat rate [10,11]. These economically crucial traits offer a holistic perspective on the necessity of integrating genetically estimated breeding values for growth rate and fat deposition traits in Yorkshire pigs with diverse genetic backgrounds.

In this study, we explored the impact of combining different breeds of Yorkshire pig populations on the precision of genomic estimated breeding values (GEBVs) for the AGE100 and BF100 traits. Our findings indicate that combining different populations can enhance GEBV accuracy under specific conditions. Bayesian models displayed superior performance compared to GBLUP and ssGBLUP models when applied to combined populations, while the latter models proved effective in predicting genomic estimated breeding values (GEBVs) within individual populations. In addition, combining the other two populations resulted in a significant third-population accuracy for the GEBV, highlighting the need for further research into the potential factors leading to this enhancement.

## 2. Materials and Methods

### 2.1. Husbandry Management of Experimental Animals

This study included 2295 pigs from three distinct regions, representing three Yorkshire pig populations: 295 Danish, 500 Canadian, and 1500 American Yorkshire pigs, and both boars and sows were selected. All data were recorded successively from 2015 to 2020. The Danish and Canadian pigs were from regions with a temperate monsoon climate, while the American pigs were from a subtropical climate. All Yorkshire pigs under investigation were in the fattening stage and resided in expansive facilities. The housing conditions maintained a feeding density ranging from 0.8 to 1.2 m^2^ per pig and with a parameter adjusted according to their respective weights. Automated feeding troughs ensured continuous access to food without interruptions. Standard disinfection procedures were applied to all the experimental animal buildings, and all the animals were vaccinated.

### 2.2. Trait-Corrected Models for Experimental Animals

Swine across all pig populations were meticulously chosen for analysis, specifically targeting individuals aged approximately 160 days and subjected to uniform feeding conditions, ensuring both health and freedom from disease. The swine underwent a standardized feeding regimen, and individual measurements were conducted at an average weight of 100 kg. Backfat thickness was assessed using ultrasound and measured during the same weighing interval between the eleventh and twelfth rib. The measurements for AGE100 were computed using the appropriate correction equations [12]: $AGE100=measured\;age+\frac{100\;kg-measured\;age}{CF}$, where CF is the correction factor (referring to the National Program for Swine Genetic Enhancement in China), $CFmale=\frac{measured\;weight}{measured\;age}\times 1.826$, and $CFfemale=\frac{measured\;weight}{measured\;age}\times 1.715$. Those for BF100 were calculated as follows [13]: $BF100male=measured\;backfat\times \frac{12.402}{12.402+0.106\times (measured\;weight-100)}$ and $BF100female=measured\;backfat\times \frac{13.705}{13.705+0.119\times (measured\;weight-100)}$.

### 2.3. DNA Extraction and Genotyping

Blood samples were transferred to sterile tubes and centrifuged at 3000 rpm for 10 min. The buffy coat layer containing leukocytes was carefully aspirated and transferred to a clean tube. Red blood cells were lysed by adding an equal volume of RBC lysis buffer (10 mM Tris-HCl, 10 mM NaCl, 1 mM EDTA, pH 8.0) and incubating at room temperature for 10 min. After the leukocyte pellet was washed with phosphate-buffered saline (PBS), cells were lysed using a commercial cell lysis buffer containing proteinase K. The lysate was incubated at 55 °C for 2–4 h to ensure complete lysis. Following the cell lysis, proteinase K was inactivated via heat treatment at 95 °C for 10 min. RNase A (20 μg/mL) was added to the lysate to degrade RNA, and the mixture was further incubated at 37 °C for 30 min. DNA was purified using a Tiangen DNA extraction kit, following the manufacturer’s instructions. Briefly, the lysate was mixed with binding buffer and transferred to a spin column. After centrifugation, the DNA was bound to the column while contaminants were removed through washing steps. Finally, purified DNA was eluted in sterile water or TE buffer.

All individuals with phenotypes were genotyped using the GeneSeek GGP porcine HD array. According to Sus scrofa version 10.2, the SNP chip consisted of 50,915 probes, and autosomes were further upgraded to the latest version of the porcine genome—Sus scrofa version 11.1. The final remaining autosomal SNPs were 34,150 Kb for the Canadian line, 34,543 Kb for the Danish line, and 34,497 Kb for the American line.

Quality control was performed through PLINK (V1.90; http://www.cog-genomics.org/; accessed on 16 March 2023). Pigs with call rates < 0.9 were excluded, and SNPs with minor allele frequencies (MAFs) below 0.05 and call rates < 0.9 were excluded in each population species.

### 2.4. Genomic Analysis of the Population

Eigenvalues and eigenvectors were acquired through PLINK (v1.90; http://www.cog-genomics.org/; accessed on 16 March 2023), with a subsequent execution of a principal component analysis (PCA) on the remaining SNPs [14]. Furthermore, linkage disequilibrium (LD, represented as r2) was computed for each population. In this investigation, SNeP software (v1.1; https://bioinformaticshome.com/tools/descriptions/SNeP.html#gsc.tab=0; accessed on 25 March 2023) was employed to determine the population effect size (Ne) [15].

### 2.5. Scenarios of Combining References

The scenarios of combining references were as follows: (1) combining all three populations (Danish, Canadian, and American lines) for prediction; (2) combining two populations to predict the third, e.g., combining the American (large-scale) and Canadian (medium-scale) lines to predict the Danish (small-scale) population or combining the American (large-scale) and Danish (small-scale) lines to predict the Canadian (medium-scale) population; and (3) predicting each population independently.

### 2.6. Statistical Analysis

#### 2.6.1. Bayesian A and Bayesian B

In 2001, Meuwissen [3] proposed the concept of GS, concurrently proposing two Bayesian models, named Bayesian A and Bayesian B, for predicting genomic estimated breeding values. The general framework for a Bayesian regression model is outlined as follows:$$y=Xb+\sum_{k=1}^{m}Z_{i}g_{i}+e,$$
where y represents the vector of phenotypes, b signifies the vector of fixed effects, including variables such as sex, X denotes the corresponding design matrix, $Z_{i}$ denotes the genotype of the i-th locus (coded as 0/1/2), $g_{i}$ represents the effect value linked to the i-th locus, and e is indicative of the random residual effect vector.

The Bayesian A model assumes that all markers contribute to the genetic effect, with the impact of genetic markers following a Gaussian distribution, $g_{i}\sim N\left(0,\sigma_{g}^{2}\right)$, while the variance adheres to an inverse chi-square distribution, $\sigma_{g_{i}}^{2}\sim \chi^{-2}\left(v,S\right)$, where v represents the degree of freedom, and S denotes the scale parameter.

Diverging from the Bayesian A approach, the Bayesian B model introduces an indicator variable represented by π, indicating the effect of the SNP. This variable operates under the assumption that most markers exhibit negligible impact (scaled by π), whereas only a limited subset of markers manifests an effect (scaled by 1−π). Furthermore, the variance associated with this specific subset of markers with an impact is characterized by an inverse chi-square distribution.
$$\left\{\begin{array}{cc} \sigma_{g_{i}}^{2}=0, & \pi \\ \sigma_{g_{i}}^{2}\sim \chi^{-2}\left(v,S\right), & 1-\pi . \end{array}\right.$$

The Bayesian A model can be conceptualized as a particular instance of Bayesian B, where the parameter π is set to 0. In this research, both the Bayesian A and Bayesian B models were applied through the utilization of the R BGLR package [16].

#### 2.6.2. BayesC

The formulation for BayesC [17] was as follows:$$y=1\mu +\sum_{i}I_{i}x_{i}s_{i}+e.$$

In this representation, y denotes a vector representing yield deviations, 1 signifies a vector of ones, μ represents the overall mean, and $x_{i}$ stands for a vector of genotypes for SNP i, incorporating −2$p_{i}$ for individuals with identical alleles, 1–2$p_{i}$ for heterozygotes, and 2–2$p_{i}$ for the alternative homozygote genotype, with pi denoting the allele frequency of the SNP i. The indicator variable $\;I_{i}$ determines whether SNP i is included in the model during the specific MCMC cycle (0/1), where the prior probability of $\;{Ι}_{i}$ being equal to 1 is denoted by π. The SNP effect $s_{i}$ is modeled to follow a normal distribution: $s_{i}∼N\left(0,\sigma_{m}^{2}\right)$. The residual e follows a normal distribution: $e∼N\left(0,D\sigma_{e}^{2}\right)$.

#### 2.6.3. GBLUP

The GBLUP model can be delineated as follows:y=μ+Xb+Zg+e,
where y represents the vector of phenotypic values, b denotes the vector of fixed effects, including variables such as sex, X stands for the corresponding design matrix, μ signifies the overall mean, Z is the design matrix establishing the link between genetic value (g) and y, and e is a vector encompassing stochastic residuals. It was hypothesized that
$$g∼N\left(0,G\sigma_{g}^{2}\right)\;and\;e∼N\left(0,I\sigma_{e}^{2}\right),$$
where $\sigma_{g}^{2}$ is the additive genetic variance, and $\sigma_{g}^{2}$ is the random residual variance. The genomic relationship matrix (G), as outlined by [18], was derived through the computation of the SNPs:$$G=\frac{\left(M-P\right){\left(M-P\right)}^{\prime }}{2\sum_{k=1}^{m}p_{k}\left(1-p_{k}\right).}$$

In this formulation, M represents an n∗m matrix, where n denotes the number of individuals, and m represents the number of SNPs; $p_{k}$ signifies the minor allele frequency of the *i*th SNP; and P is a matrix where the elements of the k-th column are 2$p_{k}$. In this investigation, GBLUP was performed through the utilization of the R qgg package [19].

#### 2.6.4. ssGBLUP

The univariate ssGBLUP model can be delineated as follows:y=Xb+Za+e.

In this framework, the incidence matrix X establishes the relationship between fixed effects b and the effect of sex, while the incidence matrix Z establishes connections between breeding values (a) and the corresponding observations in vector y, and e represents the random residual vector. It is postulated that Var(e) = $I\sigma_{e}^{2},$ where I denotes the identity matrix. The same matrix is employed in GBLUP, or a diagonal matrix incorporating weights [18]. In ssGBLUP, the breeding values exhibit a specified covariance structure in which Var(a) = $H\sigma_{e}^{2}$, with $\sigma_{e}^{2}$ denoting the genetic variance and H incorporating information from both genomic (G) and pedigree (A) relationship matrices [18,20].

### 2.7. Evaluation of the Accuracy of Genomic Prediction

In this investigation, predictive accuracy was assessed utilizing 5-fold cross-validation (CV) on actual genomic data. The genotyped individuals were randomly distributed into five nearly equitably sized subgroups. Within this framework, one subgroup was exclusively designated as the validation population, while the remaining four subgroups constituted the reference population. This cross-validation procedure was iterated five times, with each subgroup serving as the validation set once. To ensure robustness, the 5-fold CV was repeated 20 times, resulting in 20 averaged predictive accuracies.

## 3. Results

### 3.1. Assessment of Population Stratification

To analyze the genetic diversity within the three Yorkshire pig populations, we performed a principal component analysis (PCA). The outcomes revealed substantial variations in the genetic composition among Yorkshire pigs originating from the Danish, Canadian, and American lineages. In particular, PC1 was instrumental in emphasizing the genetic variations among these Yorkshire populations (Figure 1).

### 3.2. The Accuracy of the Danish (Small) Line

In this study, the Danish lines represented a smaller population within the Yorkshire pig breed, and their accuracy improved when data from the Canadian, American, and Danish lines were combined rather than relying only on the data of Danish lines to predict breeding values associated with the BF100 trait. Among the evaluated models, the ssGBLUP model demonstrated superior accuracy in predicting breeding values across combined populations. The Bayesian model consistently exhibited a superior predictive performance compared to the GBLUP and ssGBLUP models when predicting Danish lines with data derived from the Canadian and American lines, albeit without statistical significance in the observed differences. In addition, the Bayesian B model was the most accurate of the Bayesian models in relative terms (Figure 2, Table 1).

However, when examining the AGE100 trait, combining data from all three populations for predicting the Danish population resulted in greater accuracy than exclusively relying on genetic estimates from the Danish population for breeding values. Among the five models tested, no statistically significant differences were detected, although the ssGBLUP model displayed a relatively greater accuracy. In forecasting the Danish population using both the Canadian and American lines, the Bayesian models consistently surpassed the BLUP models, with the Bayesian A model demonstrating the highest overall accuracy (Figure 3, Table 2).

### 3.3. The Accuracy of the Canadian (Medium) Line

In this study, the Canadian line, representing the medium-sized population of Yorkshire pigs, showed elevated breeding values for BF100 traits when integrating data from the Danish, Canadian, and American Yorkshire pig populations rather than relying solely on the Canadian line data. The GBLUP and ssGBLUP models exhibited the highest accuracies within the combined population. When forecasting the Canadian line using data from the Danish and American lines, the Bayesian models consistently outperformed the BLUP models, although these differences lacked statistical significance. Notably, the Bayesian C model emerged as the most accurate in relative terms (Figure 4, Table 3).

Nevertheless, when considering the AGE100 trait, amalgamating data from all three populations improved the accuracy of the Bayesian models. Conversely, relying on solely data from the Canadian line proved to be more accurate than combining the three populations, especially concerning both the GBLUP and ssGBLUP models. In this scenario, the ssGBLUP model demonstrated the highest predictive accuracy. Specifically, the Bayesian C model was the most accurate when utilizing both the Danish and American lines to predict Canadian lineage populations. The Bayesian models surpassed the BLUP model in terms of accuracy (Figure 5, Table 4).

### 3.4. The Accuracy of the American (Large) Line

In this investigation, the American line, which represented the large population of Yorkshire pigs, showed that although the accuracy of the combination of the other two populations was enhanced according to the Bayesian models, the accuracy of the use of only the American data in the ssGBLUP model was low (Figure 6, Table 5).

With respect to the AGE100 trait, the predictions were more accurate when utilizing the American group alone than when combining the other two groups. The ssGBLUP model demonstrated the highest accuracy in this context (Figure 7, Table 6).

## 4. Discussion

The present study aimed to evaluate different models for the genetic estimation of breeding values and different scenarios for combined populations of Yorkshire pigs originating from distinct genetic backgrounds—namely, Danish, Canadian, and American lines. Our findings demonstrate that the amalgamation of Danish, Canadian, and American populations significantly enhanced the precision of the Bayesian model, while the BLUP model show marginal improvement in accuracy. These outcomes are consistent with our earlier findings using simulated data [6]. It has been shown that the accuracy of Bayesian variable selection models depends on the number of QTLs, with fewer QTLs resulting in higher accuracy. This trend is more evident in cross-population genomic prediction. Bayesian variable selection outperforms GBLUP when the number of QTLs is small but loses its advantage when the number of QTLs is large [21]. As an adaptation of GBLUP, several studies [7,22,23,24] have affirmed the superior performance of ssGBLUP over GBLUP. Furthermore, the outcomes of this investigation suggest that this superiority becomes more pronounced, particularly when dealing with a smaller reference population. In such instances, the relative contribution of phenotype information from non-genotyped individuals, specifically related to the selection candidates, becomes more pertinent. The ssGBLUP model had a positive effect on medium-sized and small populations of Yorkshire Pig farms, except for the AGE100 trait in the medium-sized population; in this case, the ssGBLUP model alone was more accurate at predicting the genomic estimated breeding values for the medium population itself. In addition, predicting genetic estimates of breeding values by combining large, medium, and small populations improved the prediction when only their populations were used to varying degrees for medium- and small-sized populations. Overall, the heritability estimates for the studied BF100 trait ranged from moderate to high [25], indicating a substantial genetic component and suggesting favorable responsiveness to selection. A large number of studies have shown that the ssGBLUP model is more accurate in terms of genomic selection than are other models in most cases [26,27,28,29,30]. Furthermore, the increased heritability estimates observed with the utilization of ssGBLUP compared to PBLUP may be attributed to the consideration of the Mendelian sampling term. Genomic-based methodologies, such as ssGBLUP, use the Mendelian sampling term, which pedigree-based approaches neglect. This term captures the variation among family members within a half-sib or full-sib family surrounding the family’s mean relatedness [31].

The reference population plays a pivotal role in GS, with the accuracy of GS being contingent upon both its size and its relationship to the selection of candidates. Establishing the combined reference group, comprising animals from diverse populations with distinct genetic backgrounds of the same breed, serves dual purposes: expanding the reference size and enabling the utilization of a shared reference population for genomic selection across diverse breeding populations. This strategy has been applied in various genomic selection initiatives. For example, within the Euro Genomics Coöperative U.A. initiative based in Amsterdam, the Netherlands, a unified reference population was established by amalgamating four closely affiliated populations of dairy cattle. The results indicated a substantial improvement in the reliability of genomic predictions for bulls across the four populations, showing an average increase of 10% compared to predictions based on individual reference populations alone [32]. Analogous initiatives have been pursued in beef cattle, where endeavors to form a unified reference group encompassing multiple breeds showed that prediction equations developed in multibreed populations achieved enhanced accuracy for subpopulations [33]. The enhancement of genomic prediction accuracy is facilitated through the consolidation of individuals from the same species, owing to the inherent relationship with their genealogical heritage [34]. Emre Karaman [35] discovered that in mixed populations, achieving higher accuracies in genomic evaluation is possible by amalgamating all available data from both purebred and admixed individuals. This approach outperformed other methods by considering the breed origin of alleles solely for selection candidates, along with utilizing breed-specific SNP effects estimated independently in each pure breed.

In the realm of multi-population studies, previous research has revealed the efficacy of employing individual SNP methods based on whole-genome sequencing data [36] or haplotype methods based on chip data [37,38]. These methods, leveraging either the comprehensive information provided by whole-genome sequencing or haplotype-based insights from chip data, enhance the capacity to capture LD between genetic variants and QTLs. The utilization of such approaches has been shown to effectively increase the accuracy of GP in multi-population settings. As shown in a large number of studies, high-density SNP markers and imputed data are not more accurate than medium-density markers. Therefore, this study directly used more economical and convenient 50k chips [26,39]. The implications of our findings hold significant relevance for the pragmatic aspects of Chinese pig breeding. Positioned as the primary global producer and consumer of pork, the pig industry in China is assumed to be of paramount importance in agricultural landscapes. However, a notable challenge arises from the limited or absent genetic connectedness prevalent among most Chinese pig breeding farms, stemming from infrequent genetic exchange. This deficiency in genetic interconnectedness poses a substantial hurdle, hindering the successful execution of comprehensive national or regional joint genetic evaluations thus far.

An in-depth analysis of the three Yorkshire populations in our study revealed pronounced weakness in genetic connectedness when scrutinized based on pedigree data. No shared ancestors or relatives were discerned among individuals from the Danish, Canadian, or American Yorkshire populations. Moreover, the genetic connectedness within the American Yorkshire population did not reach a sufficient level for the implementation of joint genetic evaluations, as evidenced by undisclosed data. This predicament has led to a deceleration and impediment in the genetic improvement of Chinese pig breeding, despite boasting the largest nuclear herd globally. In response to these challenges, genomic selection has emerged as a promising avenue for advancing Chinese pig breeding practices. Our results underscore the potential of combining reference populations from disparate genetic backgrounds, even in the absence of robust genetic connectedness, to significantly enhance the accuracy of genomic predictions. This insight provides a prospective solution that could propel genetic improvement in Chinese pig breeding, offering new possibilities for leveraging genomic data in the absence of extensive genetic exchange networks.

In interpreting the enhanced accuracy observed when using the 1500- and 500-head populations as the reference population and the 295-head population as the validation population, it is plausible that the larger population (1500 + 500) contributed to a broader spectrum of genetic diversity and a greater number of data points. This extended range facilitated the capture of more comprehensive genetic information, thereby elevating the prediction accuracy for smaller populations (295 heads). When incorporating a smaller population (e.g., 295 heads) into a larger reference population, there is a risk of overfitting the characteristics of the larger population, particularly if there are significant genetic differences between them. This could lead to a reduction in predictive accuracy when applied to the validation population (i.e., the 295-head population).

Conversely, when all three populations are amalgamated, even though the volume of information increases, the merger may introduce irrelevant genetic information if the populations lack genetic relatedness. This influx of extraneous genetic data may contribute to increased ‘noise’ in the model, ultimately diminishing prediction accuracy.

The Bayesian approach exhibits potential superiority over GBLUP in handling complex genetic structures, offering greater flexibility in addressing varying effect sizes of different genetic markers, especially in the presence of substantial nonadditive genetic variation. In contrast, the GBLUP approach assumes equal effects for all genetic markers, potentially lacking the required flexibility when dealing with datasets characterized by intricate genetic backgrounds. The intricate kinship relationships, lack of relatedness, and distinct genetic structures among the three populations resulted in the lower accuracies of the GBLUP and ssGBLUP models compared to those of the Bayesian approaches.

## 5. Conclusions

In summary, our investigation highlights the benefits of incorporating diverse populations into genetic models to increase the accuracy of predicting breeding values. Overall, our findings suggest that the Bayesian models outperform the GBLUP and ssGBLUP models when combining two populations to predict another, while the ssGBLUP model exhibits higher accuracy when combining three populations (American, Canadian, and Danish). Moreover, the amalgamation of two larger groups appears to significantly enhance the accuracy of the GEBV in the smaller group, although the specific mechanisms driving this improvement warrant further investigation. These findings emphasize the critical significance of incorporating population combinations within genetic models to achieve precise predictions of breeding values, especially in contexts where pig farmers contend with resource constraints.

## Figures and Tables

**Figure 1 animals-14-01098-f001:**
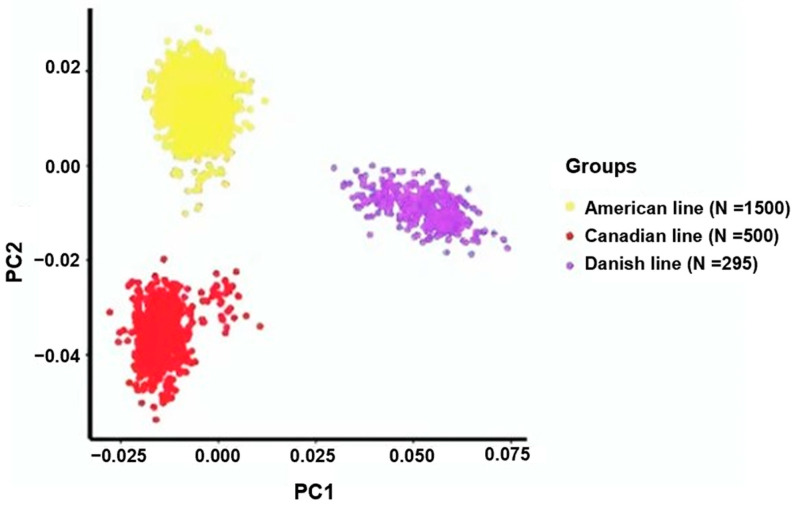
Principal component analyses revealed the association between the primary components (PC1 and PC2) and the proportion of genetic variance explained, specifically, the percentage of variation elucidated within the American, Canadian, and Danish pig lines.

**Figure 2 animals-14-01098-f002:**
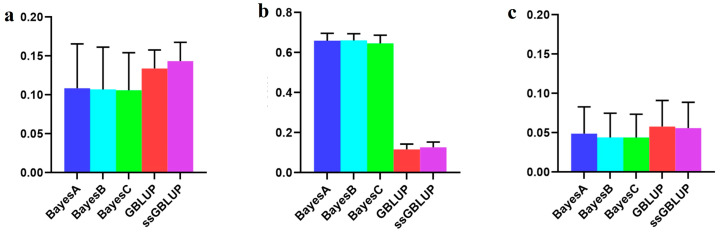
Accuracy of prediction for the BF100 trait in the Danish line. (**a**) The use of a combination of American, Canadian, and Danish populations to predict Danish populations. (**b**) The use of combined American and Canadian populations to predict Danish populations. (**c**) Individual Danish populations.

**Figure 3 animals-14-01098-f003:**
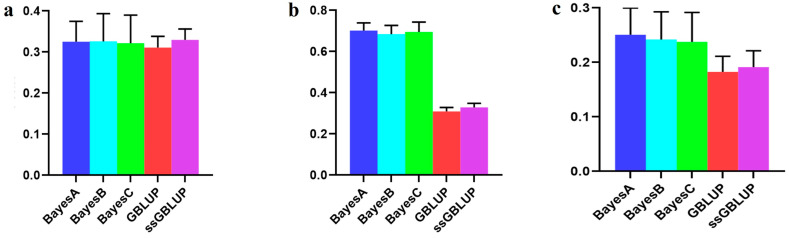
Accuracy of prediction for the AGE100 trait in the Danish cohort. (**a**) A combination of American, Canadian, and Danish populations was used to predict the prognosis of Danish populations. (**b**) The use of combined American and Canadian populations to predict Danish populations. (**c**) Individual Danish populations.

**Figure 4 animals-14-01098-f004:**
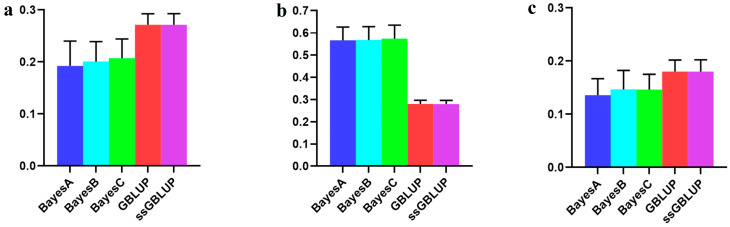
Accuracy of prediction for the BF100 trait in the Canadian line. (**a**) The use of combined American, Canadian, and Danish populations to predict Canadian populations. (**b**) The use of combined American and Danish populations to predict Canadian populations. (**c**) Individual Canadian populations.

**Figure 5 animals-14-01098-f005:**
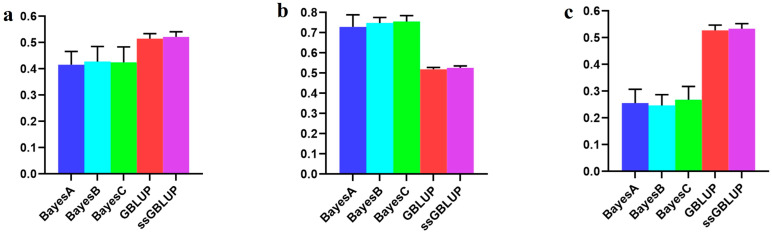
Accuracy of prediction for the AGE100 trait in the Canadian cohort. (**a**) The use of combined American, Canadian, and Danish populations to predict Canadian populations. (**b**) The use of combined American and Danish populations to predict Canadian populations. (**c**) Individual Canadian populations.

**Figure 6 animals-14-01098-f006:**
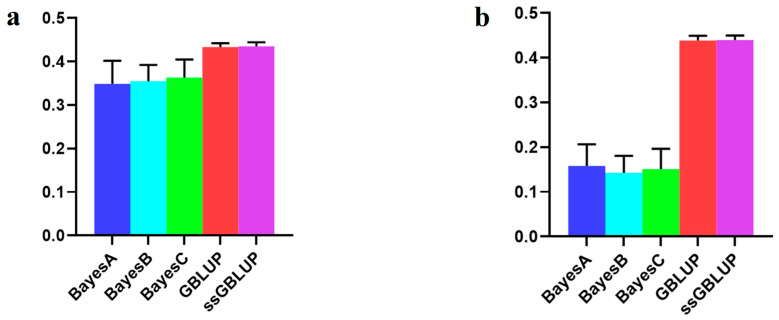
Accuracy of prediction for the BF100 trait in the American line. (**a**) A combination of American, Canadian, and Danish populations was used to predict American populations. (**b**) Individual American populations.

**Figure 7 animals-14-01098-f007:**
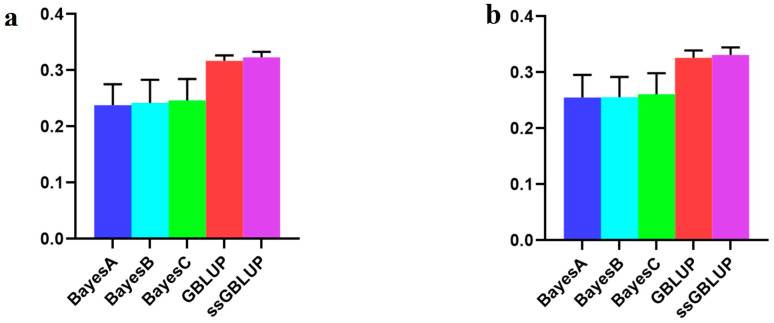
Accuracy of prediction for the AGE100 trait in the American cohort. (**a**) A combination of American, Canadian, and Danish populations was used to predict American populations. (**b**) Individual American populations.

**Table 1 animals-14-01098-t001:** Accuracy of prediction for the BF100 trait in the Danish line.

Scenarios	BayesA	BayesB	BayesC	GBLUP	ssGBLUP
Combined American, Canadian, and Danish populations	0.10850	0.10687	0.10595	0.13385	0.14336
Combined American and Canadian populations	0.65999	0.66093	0.64634	0.11604	0.12715
Danish populations individually	0.04895	0.04387	0.04385	0.05784	0.05588

**Table 2 animals-14-01098-t002:** Accuracy of prediction for the AGE100 trait in the Danish line.

Scenarios	BayesA	BayesB	BayesC	GBLUP	ssGBLUP
Combined American, Canadian, and Danish populations	0.32526	0.32599	0.32159	0.31096	0.32953
Combined American and Canadian populations	0.70134	0.68404	0.69489	0.30901	0.32876
Danish populations individually	0.25057	0.24194	0.23761	0.18266	0.19131

**Table 3 animals-14-01098-t003:** Accuracy of prediction for the BF100 trait in the Canadian line.

Scenarios	BayesA	BayesB	BayesC	GBLUP	ssGBLUP
Combined American, Canadian, and Danish populations	0.19236	0.20061	0.20714	0.27130	0.27133
Combined American and Danish populations	0.56656	0.56813	0.57448	0.28029	0.27973
Canadian population individually	0.13579	0.14652	0.14622	0.18004	0.18012

**Table 4 animals-14-01098-t004:** Accuracy of prediction for the AGE100 trait in the Canadian line.

Scenarios	BayesA	BayesB	BayesC	GBLUP	ssGBLUP
Combined American, Canadian, and Danish	0.41599	0.42777	0.42452	0.51505	0.52175
Combined American and Danish populations	0.72887	0.74858	0.75569	0.51873	0.52609
Canadian population individually	0.25572	0.24653	0.26800	0.52769	0.53372

**Table 5 animals-14-01098-t005:** Accuracy of prediction for the BF100 trait in the American line.

Scenarios	BayesA	BayesB	BayesC	GBLUP	ssGBLUP
Combined American, Canadian, and Danish populations	0.34910	0.35499	0.36329	0.43341	0.43526
American population individually	0.15793	0.14229	0.15109	0.43869	0.43978

**Table 6 animals-14-01098-t006:** Accuracy of prediction for the AGE100 trait in the American line.

Scenarios	BayesA	BayesB	BayesC	GBLUP	ssGBLUP
Combined American, Canadian, and Danish populations	0.23780	0.24182	0.24619	0.31689	0.32331
American population individually	0.25502	0.25556	0.26096	0.32570	0.33130

## Data Availability

Data are contained within the article.

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
