# Peer review of "Genomic Prediction of Growth Traits in Yorkshire Pigs of Different Reference Group Sizes Using Different Estimated Breeding Value Models"

_animals, 2024, doi:10.3390/ani14071098_

Round 1

Reviewer 1 Report

Comments and Suggestions for Authors

Authors have taken a good effort to organise this study, analyse the data and draft the manuscript. I have a few suggestions and queries which have been listed below.        

Major comments:

1.      Materials and methods: the authors could also mention about the differences in Yorkshire genotypes based on the differences in their region/farms/cities. Also does these three region experience different climatic conditions? Were all the pigs under each genotype, Danish, Canadian and American, selected from a single farm (thereby making the total number of farms to three)?

2.      Subheading 2.2. Animals and phenotypes: Kindly indicating the month/year of trait recording

3.      Subheading 2.2. Animals and phenotypes: Kindly use the multiplication symbol (×) instead of asterisk (*) while indicating multiplication in the trait computation

4.      Materials and methods: Kindly indicate a few statements on the sample collection for DNA extraction and the genotyping. The authors have briefed this in the abstract however a description in the materials and methods section would be ideal too.

5.      Subheading 2.4 Statistical analysis:

a.       In all the five statistical models, the authors indicated sex as the fixed effect. If so, kindly mention under the subheading 2.1 that both boars and sows were selected.

b.      Didn’t the authors include other fixed effects like breed-type (Danish, Canadian and American) or were these models run separately for each breed-type

6.      The authors have taken a good effort to draft the materials and methods section, however I feel it can be improved. In the abstract the authors have briefed about the methodology (for instance, assessing the accuracy of GS by using three approaches) which however was not reflected in the materials and methods. Maybe this was done to avoid repetition, however incorporating such details elaborately in the materials and methods section will help readers understand the approach and if need be, adopt it for their research.

7.      Conclusion: Line 421-424: Would it be appropriate to make a statement that Bayesian models are superior than GBLUP and ssGBLUP when the later had higher prediction values for most trait and breed type models? I agree with the author’s earlier statement in the discussion that amalgamation of Danish, Canadian, and American populations significantly enhanced the precision of the Bayesian model, while the BLUP model showed marginal improvement in accuracy. Having said so, the prediction values were still higher in GBLUP and ssGBULP. Hence kindly reconsider editing this statement.

8.      Would the differences in sample size be an a

9.      For some traits and breed lines (like AGE100 for Canadian line and BF100 for American line) the accuracy of prediction was better for the individual line (GBULP and ssGBULP) than the amalgamation with other lines. What could be the probable justification for such result?

a.       Likewise, in such cases there were marginal differences in accuracy of prediction in GBULP and ssGBULP. Hence wouldn’t this be a better prediction model?

Minor comments:

1.      Line 16 and line 17: Kindly rectify the typing error “fo-cusing” as “focusing” and “accura-cy” as “accuracy”. Kindly rectify such typing errors throughout the text

2.      Line 241: the term NAFLD seems to be appearing for the first time in the text, kindly expand it.

3.      In certain portions in the text the authors have mentioned BLUP and not GBULP or ssGBULP. Was this done intentionally or was it a typing error?

Author Response

Response to the reviewer’s comments on our manuscript:

We appreciate the valuable comments received from section editor and the reviewer. They have helped us to significantly improve our study.

Reviewers' Comments:

1)Materials and methods: the authors could also mention about the differences in Yorkshire genotypes based on the differences in their region/farms/cities. Also does these three region experience different climatic conditions? Were all the pigs under each genotype, Danish, Canadian and American, selected from a single farm (thereby making the total number of farms to three)?

Response:

Thanks for precious suggestion. We have mentioned these Yorkshire were from three different cities (Line 37 – line 39), and added these three different regions’ climates (Line 99 – line 103).

2)Subheading 2.2. Animals and phenotypes: Kindly indicating the month/year of trait recording

Response:

Thanks for precious suggestion. All data were recorded successively from 2015 to 2020, added as suggestion (Line 101).

3) Subheading 2.2. Animals and phenotypes: Kindly use the multiplication symbol (×) instead of asterisk (*) while indicating multiplication in the trait computation Response:

Thanks for precious suggestion. Revised as suggested (Line117 – line 121).

4)Materials and methods: Kindly indicate a few statements on the sample collection for DNA extraction and the genotyping. The authors have briefed this in the abstract however a description in the materials and methods section would be ideal too. Response:

Thanks for precious suggestion. Added as suggestion (Line 123 – Line 146).

5)Subheading 2.4 Statistical analysis:

In all the five statistical models, the authors indicated sex as the fixed effect. If so, kindly mention under the subheading 2.1 that both boars and sows were selected.

Response:

Thanks for precious suggestion. Added as suggested (Line100 – line 101).

  1. Didn’t the authors include other fixed effects like breed-type (Danish, Canadian and American) or were these models run separately for each breed-type

Response:

Thanks for precious question. This experiment was more concerned with the effect of combining Yorkshire pigs of different genetic backgrounds of the same breed on the accuracy of GEBV and explored the experimental results mainly through different combining scenarios, so no other effects were added to the models.

6)The authors have taken a good effort to draft the materials and methods section, however I feel it can be improved. In the abstract the authors have briefed about the methodology (for instance, assessing the accuracy of GS by using three approaches) which however was not reflected in the materials and methods. Maybe this was done to avoid repetition, however incorporating such details elaborately in the materials and methods section will help readers understand the approach and if need be, adopt it for their research.

Response:

Thanks for precious suggestion. Added as suggested (Line 154 – Line 159)

7)Conclusion: Line 421-424: Would it be appropriate to make a statement that Bayesian models are superior to GBLUP and ssGBLUP when the later had higher prediction values for most trait and breed type models? I agree with the author’s earlier statement in the discussion that amalgamation of Danish, Canadian, and American populations significantly enhanced the precision of the Bayesian model, while the BLUP model showed marginal improvement in accuracy. Having said so, the prediction values were still higher in GBLUP and ssGBULP. Hence kindly reconsider editing this statement.

Response:

Thanks for precious suggestion. Revised as suggested (Line 436 – Line 440).

8)Would the differences in sample size be an a

Response:

Sorry, we can’t get the point.

9)For some traits and breed lines (like AGE100 for Canadian line and BF100 for American line) the accuracy of prediction was better for the individual line (GBULP and ssGBULP) than the amalgamation with other lines. What could be the probable justification for such result?

Likewise, in such cases there were marginal differences in accuracy of prediction in GBULP and ssGBULP. Hence wouldn’t this be a better prediction model?

Response:

Thanks for precious question. As we were working with chip data, different traits and different populations may lead to differences in the accuracy of different GEBV models due to differences in genetic structure. Different GEBV models in different populations may capture different patterns of genetic structure and linkage disequilibrium in different breeds, thus affecting accuracy. The original manuscript summary section was inaccurately described and has now been corrected.

Minor comments:

1)Line 16 and line 17: Kindly rectify the typing error “fo-cusing” as “focusing” and “accura-cy” as “accuracy”. Kindly rectify such typing errors throughout the text Response:

Thank you so much for pointing the mistake. Revised all as suggested.

2)Line 241: the term NAFLD seems to be appearing for the first time in the text, kindly expand it.

Response:

Thank you so much for pointing the mistake. In fact, it’s an error, we have deleted it.

3)In certain portions in the text the authors have mentioned BLUP and not GBULP or ssGBULP. Was this done intentionally or was it a typing error?

Response:

Thanks for mentioning it. We are so sorry about our careless and have revised all the mistake.

Reviewer 2 Report

Comments and Suggestions for Authors

The authors evaluated different models to predict breeding values for growth efficiency and lean meat percentage in three populations of Yorkshire pigs of differing numbers. This paper is interesting. Title of the manuscript is adequate to the its text. The investigations were done on sufficient animal material. All presented figures and tables are necessary. The manuscript is well-written, however, there are several areas require attention. Were the pigs kept on one farm or in different locations? Did body weight and backfat thickness measurements in the same time ?

Specific comments:

1)      Please revise the abstract and clearly indicate the purpose of the work.

2)      L 56 „GS” Please define the abbreviation in the introduction section.

3)      L 88-105 Please delete the entire paragraph; This information relates to other sections. At the end of the Introduction section, indicate clearly the purpose of the work.

4)      L 95 GEBV Please explain the abbreviation.

5)      L 102 GEBVs Please explain the abbreviation.

6)      L 108 “three discernible breeds:” rather “ three Yorkshire pig populations”

7)      L 229 ”within the Large White pig breed” ? Is the name breed correct ?

8)      L 260 “…Large White pigs…” ? or Yorkshire

9)      L 292 “Large White pigs” ?

10)  L 327 “Large White Pig farms” ?

11)  L 318 “with fewer QTLs resulting in higher accuracy” Please explain the reason.

12)  L 327 “Large White Pig” ?

13)  L333 „from moderate to high” Please cite the reference for this sentence; The authors write about heritability.

14)  L 335 The authors write that a large number of studies and cite only 3 studies.

15)  L 337 Why the authors write about heritability? If the studies calculated heritability then please provide h2 values in the tables.

16)  Please adapt the references to the requirements of the Animals journal.

Author Response

Response to the reviewer’s comments on our manuscript:

We appreciate the valuable comments received from section editor and the reviewer. They have helped us to significantly improve our study.

Reviewer:

The authors evaluated different models to predict breeding values for growth efficiency and lean meat percentage in three populations of Yorkshire pigs of differing numbers. This paper is interesting. Title of the manuscript is adequate to the its text. The investigations were done on sufficient animal material. All presented figures and tables are necessary. The manuscript is well-written, however, there are several areas require attention. 

Were the pigs kept on one farm or in different locations? Did body weight and backfat thickness measurements in the same time?

Response:

Thanks for precious suggestion. We have mentioned these Yorkshire were from three different cities (Line 37 – line 39, Line 99 – Line 101 ). The traits of AGE100 and BF100 were measured simultaneously in this experiment. BF trait was measured by ultrasound.

Specific comments:

  • Please revise the abstract and clearly indicate the purpose of the work.

Response:

Thanks for precious suggestion. Revised as suggested.

2)      L 56 „GS” Please define the abbreviation in the introduction section.

Response:

Thanks for precious suggestion. Added as suggestion (Line55).

3)      L 88-105 Please delete the entire paragraph; This information relates to other sections. At the end of the Introduction section, indicate clearly the purpose of the work.

Response:

Thanks for precious suggestion. Revised as suggestion (Line 88 – Line 96).

4)      L 95 GEBV Please explain the abbreviation.

Response:

Thanks for precious suggestion. Added as suggestion (Line 89).

5)      L 102 GEBVs Please explain the abbreviation.

Response:

Thanks for precious suggestion. Added as suggestion (Line 94).

6)      L 108 “three discernible breeds:” rather “ three Yorkshire pig populations” Response:

Thanks for precious suggestion. Revised as suggestion (Line 99).

7)      L 229 ”within the Large White pig breed” ? Is the name breed correct ?

Response:

Thanks for precious suggestion. Revised as suggestion.

8)      L 260 “…Large White pigs…” ? or Yorkshire

Response:

Thanks for precious suggestion. Revised as suggestion (Line 244).

9)      L 292 “Large White pigs” ?

Response:

Thanks for precious suggestion. Revised as suggestion (Line 276).

10)  L 327 “Large White Pig farms” ?

Response:

Thanks for precious suggestion. Revised as suggestion.

11)  L 318 “with fewer QTLs resulting in higher accuracy” Please explain the reason.

Response:

Thanks for the precious question. The GBLUP model assumes an infinitesimal effect for each SNP, contributing equally to the variation. In contrast, Bayesian variable selection models assign significant effects to a few SNPs strongly associated with the trait, while the rest have minimal effects. When the number of QTLs is small relative to the number of independent chromosome segments (), Bayesian models excel by focusing on a select few SNPs. However, with a higher number of QTLs, each SNP has a minor impact, and no distinct subset emerges for significant effects. In such cases, the Bayesian model's advantage over GBLUP lessens, as both effectively estimate the same number of effects. Posterior variance assignment in Bayesian models equates to the GBLUP's infinitesimal effect assumption. to the assumption of the infinitesimal model underlying GBLUP.

Cited: van den Berg S, Calus M P L, Meuwissen T H E, et al. Across population genomic prediction scenarios in which Bayesian variable selection outperforms GBLUP[J]. BMC genetics, 2015, 16: 1-12.

12)  L 327 “Large White Pig” ?

Response:

Thanks for precious suggestion. Revised as suggestion, all the “Large White” have revised to “Yorkshire”.

13)  L333 „from moderate to high” Please cite the reference for this sentence; The authors write about heritability.

Response:

Thanks for precious suggestion. Revised as suggestion (Line 349).

14)  L 335 The authors write that a large number of studies and cite only 3 studies.

Response:

Thanks for precious suggestion. Added as suggestion.

15)  L 337 Why the authors write about heritability? If the studies calculated heritability then please provide h2 values in the tables.

Response:

Thanks for precious suggestion. Using genetic variance and total variance, we calculated the heritability of the AGE100 trait for the three populations combined to be 0.46, and the heritability of the BF100 trait to be 0.68. In conjunction with several studies in the literature on the heritability of these two traits in Yorkshire pigs, we learned that these two traits are of medium to high heritability.

16)  Please adapt the references to the requirements of the Animals journal.

Response:

Thanks for precious suggestion. Revised as suggestion.

Reviewer 3 Report

Comments and Suggestions for Authors

After recovery from the ASF outbreak and rebuilding the Chinese pig industry, several new facilities and different sources of swine genetics were introduced into this sector.  However, it helps to reestablish pork production in China. There remain many questions: How can they implement a united pig improvement program, and how can they use pigs from different origins (diverse selection backgrounds) from the same breed for genetic progress in the country?

The authors used genomic selection to improve the accuracy of genomic prediction for multi-populations in pigs. This had been previously tested in a simulation study that examined various scenarios to assess the effect of heritability and the merging of different populations on the accuracy of GEBVs (Same authors, 2024).    

Lines 34-45 in the Abstract and 88-98 in the Introduction are the same description, which is obligatory in the summary but not necessary in the introduction in detail.  Please rephrase 88-98.

The titles of the first two subchapters in Materials and Methods should be corrected because they do not agree with their contents. What do you mean by dietary inventory? What are the phenotypes?  

The first subchapter, technological description, is very general, and some of this information is irrelevant. Each genotype has its own feeding manual for reaching its genetic potential during growing and finishing. However, in the next subchapter, it is written that they were subjected to uniform feeding conditions. What does this mean? The measurement of the two parameters (investigated traits) should be described in detail.  

The statistical models seem to be adequate and transparent.

Results demonstrated that the three populations are genetically distinct. The small population (Danish) is more vulnerable to genetic selection; however, using the medium and large population data, GEBV was more accurate using Bayesian models compared to its own data, GBLUP and ssGBLUP. The same trend was found in the Canadian population but with higher accuracy. However, its own data for AGE100 was superficial for prediction by GBLUP and ssGBLUP. Estimating the USA population in any combination resulted in the best GBLUP and ssGBLUP.

The discussion is well-developed and supported by the results and the final conclusions.

However, it should be kept in mind that the investigated/estimated parameters have a medium-high heritability; thus, some of the reproductive traits could have totally different figures.

The references should be proofread.

Author Response

Response to the reviewer’s comments on our manuscript:

We appreciate the valuable comments received from section editor and the reviewer. They have helped us to significantly improve our study.

Reviewer:

After recovery from the ASF outbreak and rebuilding the Chinese pig industry, several new facilities and different sources of swine genetics were introduced into this sector.  However, it helps to reestablish pork production in China. There remain many questions: How can they implement a united pig improvement program, and how can they use pigs from different origins (diverse selection backgrounds) from the same breed for genetic progress in the country?

The authors used genomic selection to improve the accuracy of genomic prediction for multi-populations in pigs. This had been previously tested in a simulation study that examined various scenarios to assess the effect of heritability and the merging of different populations on the accuracy of GEBVs (Same authors, 2024).    

  1. Lines 34-45 in the Abstract and 88-98 in the Introduction are the same description, which is obligatory in the summary but not necessary in the introduction in detail.  Please rephrase 88-98.

Response:

Thanks for precious suggestion. Rephrased as suggestion (Line 88 – Line 96).

  1. The titles of the first two subchapters in Materials and Methods should be corrected because they do not agree with their contents. What do you mean by dietary inventory? What are the phenotypes?  

Response:

Thanks for precious suggestion. Revised as suggestion (Line 98, Line 118).

  1. The first subchapter, technological description, is very general, and some of this information is irrelevant. Each genotype has its own feeding manual for reaching its genetic potential during growing and finishing. However, in the next subchapter, it is written that they were subjected to uniform feeding conditions. What does this mean? The measurement of the two parameters (investigated traits) should be described in detail.

Response:

Thanks for precious suggestion. We have removed irrelevant content and described methods for measuring traits (Line 99 – Line 115).

The statistical models seem to be adequate and transparent.

  1. Results demonstrated that the three populations are genetically distinct. The small population (Danish) is more vulnerable to genetic selection; however, using the medium and large population data, GEBV was more accurate using Bayesian models compared to its own data, GBLUP and ssGBLUP. The same trend was found in the Canadian population but with higher accuracy. However, its own data for AGE100 was superficial for prediction by GBLUP and ssGBLUP. Estimating the USA population in any combination resulted in the best GBLUP and ssGBLUP.

The discussion is well-developed and supported by the results and the final conclusions.

Response:

Thanks for enjoy this manuscript. We appreciate it.

  1. However, it should be kept in mind that the investigated/estimated parameters have a medium-high heritability; thus, some of the reproductive traits could have totally different figures.

Response:

Thanks for bring this up, we understand that the probability is that these results will be very different in reproductive traits.

  1. The references should be proofread.

Response:

Thanks. It's been proofread.